# Who Knows? Information Received, and Knowledge about, Cancer, Treatment and Late Effects in a National Cohort of Long-Term Childhood, Adolescent and Young Adult Cancer Survivors

**DOI:** 10.3390/cancers14061534

**Published:** 2022-03-16

**Authors:** Micol E. Gianinazzi, Cecilie E. Kiserud, Ellen Ruud, Hanne C. Lie

**Affiliations:** 1National Advisory Unit on Late Effects after Cancer Treatment, Oslo University Hospital, 0379 Oslo, Norway; migian@ous-hf.no (M.E.G.); ckk@ous-hf.no (C.E.K.); 2Department of Pediatric Medicine, Oslo University Hospital, Rikshospitalet, 0372 Oslo, Norway; elruud@ous-hf.no; 3Institute of Clinical Medicine, University of Oslo, 0372 Oslo, Norway; 4Department of Behavioral Medicine, Institute of Basic Medical Sciences, Faculty of Medicine, University of Oslo, 0372 Oslo, Norway

**Keywords:** CCS, YACS, information provision, late effects, follow-up care, cancer survivorship

## Abstract

**Simple Summary:**

With the growing population of cancer survivors, survivorship management has become central for both medical professionals and patients. This entails, among other factors, empowering survivors with the necessary knowledge about their medical history and their risk for late effects, because informed patients make better lifestyle and health choices. Although a lack of information and low satisfaction with information received are fairly well-documented phenomena among childhood cancer survivors, less is known about survivors of young adult cancer and populations of long-term survivors no longer engaged in follow-up care. This paper aims to fill this gap by investigating information provision in childhood, adolescent and young adult cancer survivors.

**Abstract:**

Background: Knowledge of medical history and late effects is central in modern survivorship management, especially for long-term childhood, adolescent and young adult cancer survivors’ (CAYACS) with long life expectancy rates and high risks of late effects. Identifying information and knowledge gaps is, therefore, important. As part of the population-based NOR-CAYACS study, we investigated the following: (1) written information received about their disease and treatment, and any information about late effects; (2) satisfaction with this information and associated factors; (3) knowledge about late effects and factors associated with low knowledge of specific late effects. Material and methods: A questionnaire-based survey (Nor-CAYACS) was mailed to 5361 CAYACS (childhood cancers, breast and colorectal cancer, acute lymphatic leukemia, non-Hodgkin lymphoma and malignant melanoma) identified by the Cancer Registry of Norway (CRN). Of these, 2018 answered questions about disease and late effects information and knowledge. Exposure variables were extracted from the questionnaire and CRN. Unfortunately, it was not possible to stratify by treatment in the analyses. We ran descriptive statistics for comparisons and logistic regressions to identify factors associated with outcomes of interest. Results: Overall, 50% to 60% of survivors reported not having received written information about their disease and treatment, or any information about late effects. There was a large variation in reported knowledge across 17 late effects. Lower levels of knowledge were associated with male sex, lower education and poorer health literacy in multivariable regression models. Conclusions: Knowledge of cancer history and risks of late effects is essential for effective self-management, yet significant information and knowledge gaps were reported in this population-based sample of long-term CAYACS. Systematic approaches to making (up-to-date) information available to long-term survivors are needed to ensure that information does not get lost in medical and life transitions.

## 1. Introduction

With the number of cancer survivors steadily increasing due to improvements in diagnostics and treatment [1,2], survivorship management has become central for both clinicians and patients [3,4,5,6,7,8]. Given that the cancer treatment place survivors at risk of increased morbidity (late effects) and early mortality compared to the general population [4,5,8,9,10], long-term follow-up care is recommended [11]. Additionally, empowering survivors with knowledge about their medical history and their risk for late effects is important, as informed patients make better lifestyle and health choices [3,12,13]. Lack of knowledge about the personal medical history has been linked to higher loss to follow-up care, higher costs at the public health level and more depression and anxiety among the cancer survivors [4,14,15]. Educating childhood cancer survivors (CCS) and young adult cancer survivors (YACS) is of particular importance because of their young age at diagnosis, long life expectancy and relatively high risk of late effects [4,5,16,17]. For children, it is especially important to ensure that information reaches the survivors themselves, not only their parents, and that information does not get lost in transition from pediatric to adult care [18,19,20]. YACS represents a group with unique needs [8], as their cancer disease and survivorship coincides with educational attainment, employment and starting their own families. They are also more prone to loss to follow-up and adhere less to medical recommendations than older survivors [21]. Several studies have shown a general lack of information received about the cancer diagnosis, treatment and risk for late effects among childhood, adolescent and young adult cancer survivors (CAYACS) [12,21,22,23]. Similarly, important knowledge gaps among these groups of cancer survivors are well documented [24,25,26,27]. In addition, low satisfaction with the information received and a need for more information have been reported [12,21,28,29]. Less is known, however, regarding information received and knowledge about specific late effects [30,31]. This is especially true for YACS, which historically have not been considered as a distinct group of survivors with specific needs and preferences [32].

Lower age at diagnosis and cancer type (renal tumor) have been associated with lower disease and late effects knowledge among CCS [30]. Further, the perceived need for more information has been linked to anxiety and depression, as well as lower health-related quality of life [12]. Lastly, health literacy—an individual’s ability to access, process, critically evaluate and act upon health information [33,34,35]—is likely a relevant predictor of satisfaction with information received and cancer history knowledge, but remains largely unexplored among CAYACS [36,37,38]. 

The present study is part of a larger national study on life after cancer in childhood, adolescent and young adult age, the NOR-CAYACS study [39]. Here, we investigate (1) whether long term CCSs and YACSs report having received written information on diagnosis and treatment, and any information about late effects; (2) survivors’ satisfaction with the information received and associated factors; and (3) survivors’ reported knowledge about late effects and factors associated with low knowledge about selected late effects, i.e., fatigue, second cancers, hormonal changes and cognitive impairment. 

## 2. Materials and Methods

### 2.1. Participants and Procedures

We used data from the NOR-CAYACS study [39] [30]. Eligible participants were identified through the Cancer Registry of Norway (CRN) [40]. 

Study inclusion criteria were: survivors aged ≥18 years at the time of the study, diagnosed between 1985 and 2009, with a minimum of five years since the initial cancer diagnosis of any childhood cancer (CC, excluding central nervous system tumors) at ages 0–18 years; or breast cancer (BC, ICD-10: C50, stages < III), colorectal cancer (CRC, ICD-10: C18-20), non-Hodgkin lymphoma (NHL, ICD-10: C82-85), leukemia (LEUK, ICD-10: C91-96) or a random subsample of malignant melanomas (MM, ICD-10: 43), diagnosed at ages 19–39 years [39,41]. Other diagnostic groups such as testicular, ovarian and Hodgkin’s lymphoma were not included due to inclusion in concurrently running studies at our department.

The questionnaire was sent by mail to eligible survivors in 2015–2016 and covered the following topics: sociodemographic background, experienced late effects, health care use and needs, information provision, work ability and financial burden, physical health, mental health, fatigue, lifestyle and health literacy [39].

### 2.2. Outcome Variables from the Questionnaire

#### 2.2.1. Reported Information Received

We assessed the reported information received in three separate questions. Have you ever received written information about the cancer? Have you ever received written information about the treatment you received? Have you ever received any information about potential late effects after cancer treatment? Participants could answer “yes”,” no” or “I don’t know”.

#### 2.2.2. Satisfaction about Information Received on Late Effects

If they had received information about late effects, they were asked to indicate the source of information from the following: medical specialist at hospital, general practitioner, cancer nurse at hospital, cancer nurse at municipality, cancer society or other patient organization. They were also asked to indicate how satisfied they were with the information received: “very satisfied”, “satisfied” or “not satisfied”. For the logistic regression, we created a binomial variable merging the two categories “very satisfied” and “satisfied” vs. the “non-satisfied” group. 

### 2.3. Knowledge about Late Effects

We asked survivors if they were aware that cancer treatment could result in certain late effects such as hormonal changes, reduced fertility, cardiovascular disease, lung problems, fatigue, dental problems, cognitive impairments, hearing problems, muscle cramps, neurological pain, numbness in hands or feet, new cancers, changed sexual function, osteoporosis, lymphedema and psychological reactions, as well as radiation damage to skin, muscles and tissues. Response options were “I know about it”, “I have experienced it myself”, or “I don’t know about it”. We combined the first two responses to represent knowledge of a certain late effect.

### 2.4. Exposure Variables from the Questionnaire

We assessed health-related quality of life (HRQoL) using the Short-Form 12 (SF-12) survey. The SF-12 yields two summary scores: the physical component summary (PCS) and mental component summary (MCS). Scores were T-standardized with a mean of 50 and standard deviation of 10. Higher scores represent better HRQoL. The SF-12 showed good retest reliability (between 0.76 and 0.89) and validity in CCS [42,43].

Health literacy [33,44] was assessed using the 16-item health care subscale of the European Health Literacy Survey Questionnaire (HLS-EU-Q [33]), translated into Norwegian according to standard procedure. The HLS-HC (Health Literacy Survey—Healthcare Subscale) assesses the perceived difficulty of health information processing in a health care context. Using items in the format “on a scale from very easy to very difficult, how easy would you say it is to…?”, responses are elicited on a four-point Likert scale distinguishing between the categories “very difficult”, “fairly difficult”, “fairly easy” and “very easy”. Following HLS-EU instructions [45], sum scores between 0 and 50 were calculated, with a higher score indicating better health literacy. 

Psychological distress was assessed using the Patient Health Questionnaire-9 (PHQ-9) for depression, which scores each of the nine DSM-IV criteria as “0” (not at all) to “3” (nearly every day), with scores then summed up, with a higher total score indicating higher depression symptom severity [46]. We used the 7-item Hospital Anxiety and Depression Scale—Anxiety Subscale (HADS-A), which has good validity [47]. The item scores ranged from 0 (not present) to 3 (highly present), with sum scores ranging from 0 to 21, where a higher score indicates higher anxiety levels. 

Educational achievement was assessed by asking about the highest education ever achieved. We categorized education into mandatory school or less (<10 years), high school (10–13 years) and university college or university (>13 years).

The number of self-reported late effects was measured by adding the scores for the 17 late effects described above to which survivors responded, “I have experienced myself”. The sum scores were divided into four categories: no late effects, 1–2 late effects, 3–4 late effects and ≥5 late effects.

Participants were asked to indicate the types of cancer treatment they had received. We categorized these treatments into minimal surgery (melanoma survivors with surgery only), local treatment (surgery or radiation only), systemic treatment only (chemotherapy only) or multiple treatments (any combination of treatment modalities) [41]. 

### 2.5. Exposure Variables from the CRN

For the analyses, we classified diagnoses into CC, BC, CRC, NHL, LEUK and MM. We further generated a variable for type of survivor (CCS vs. YACS). Gender, date of birth and date of the cancer diagnosis were also provided by the CRN. We calculated the age at study as the time from birth to 15 May 2015 and time since diagnosis as time from first cancer diagnosis to 15 May 2015. The number of cancer diseases was also extracted from the CRN and categorized into 1, 2 and 3 or more. 

### 2.6. Statistical Analyses

We performed all statistical analyses using Stata 17.0 (Stata Corporation, College Station, TX, USA). A *p*-value < 0.05 was considered statistically significant if not other specified. For descriptive statistics and aims 1, 2a and 3a, we used numbers, proportions and chi-squared tests for categorical variables and means with standard deviations and *t*-tests for continuous variables. To investigate factors associated with low satisfaction with information about late effects (aim 2b), we included only survivors who reported to have received information about late effects from at least one provider. Within this sample, we performed univariable and multivariable logistic regression models with low satisfaction as the outcome, and including all variables of interest based on the literature and expert knowledge. We did not include cancer diagnosis in the multivariable model due to collinearity with type of survivor and included only 2 of the 4 variables related to age and time. To investigate whether the factors associated with low satisfaction differed by type of survivor, we included an interaction term with each covariate and type of survivor in the multivariable model. We considered a *p*-value < 0.01 as statistically significant for the interaction to account for multiple testing. To investigate factors associated with low knowledge about late effects (aim 3b), we performed univariable and multivariable logistic regression models. For the outcome of the model, we grouped the categories “I know about it” and “I know because I experienced it myself” into one category, “has knowledge”, and compared it with the category “does not have knowledge”. We ran individual models for each of the outcomes low knowledge about fatigue, second cancers, hormonal changes and cognitive impairment. For each of these outcomes, we ran the same univariable and multivariable models as described above for the models on satisfaction.

### 2.7. Ethical Considerations

The study was granted concession by The Norwegian Data Protection Authority (15/00395-2/CGN), and approved by the Regional Committee for Medical Research Ethics (2015/232 REK Sør-Øst B) and the Data Protection Officer at Oslo University Hospital and the CRN.

## 3. Results

### Characteristics of the Study Population

Of the eligible 5361 survivors that were contacted, 2104 (39.3%) returned the questionnaire (Appendix A). For the current analysis, we included 2018 survivors, 607 (30.1%) CCS and 1411 (69.9%) YACS, while 86 survivors had to be left out because they did not meet the inclusion criteria (at least 5 years since the last cancer). The most frequent CC diagnoses were leukemia (32%) and lymphoma (28%), while BC (39%) and MM (22%) were the most frequent YAC diagnoses. In comparison with YACS respondents, more CCS were males (43% vs. 27%, *p* < 0.001), were younger at the time of study (mean 30.2 years vs. 48.9, *p* < 0.001) and had a longer time since diagnosis (mean 19.1 years vs. 15.4, *p* < 0.001; Table 1). 

Aim 1: Information received on diagnosis, treatment and late effects.

Overall, 40–60% of the survivors reported not to have received written information about their cancer disease or treatment, or any information about potential late effects (Figure 1a,b).

Figure 1 shows the reported written information ever received on disease, treatment and late effects, stratified by type of cancer survivor (CC vs. YAC survivors; Figure 1a) and type of treatment (Figure 1b). The *p*-values were calculated using Chi-square statistics.

More YACS than CCS reported not having received written information on cancer disease (53% vs. 42%, *p* < 0.001) or on treatment (64% vs. 50%, *p* < 0.001) or any information about late effects (54% vs. 41%, *p* > 0.001; Figure 1a). 

Overall, the proportions of survivors who reported to have received written information about disease and treatment and any information about late effects ranged between 20% and 40%. Survivors who were treated with systemic or multiple treatments reported more often to have received such information on disease, treatment and late effects than those with minimal or local treatment (all *p*-values < 0.001; Figure 1b). 

Aim 2a: Satisfaction with the information received about late effects stratified by source of information.

YACS and CCS responses were similar regarding satisfaction with the information received on late effects (0.061 < *p*-value < 0.566). Survivors were most satisfied (very satisfied and satisfied) with the information received by specialist physicians at the hospital (67% CCS and 75% YACS) and cancer nurses at the hospital (70% CCS and 66% YACS). They were least satisfied with information provided by municipality cancer nurses (30% CCS and 40% YACS) followed by general practitioners (53% CCS and 41% YACS). 

Aim 2b: Factors associated with low satisfaction with information received about late effects.

In the univariable models, female sex (odds ratio (OR) = 0.44 comparing males to females, 95% confidence interval (CI) 0.29–0.65, *p* < 0.001), lower health literacy levels (OR= 0.94, 95% CI 0.92–0.97, *p* < 0.001), higher anxiety (OR = 1.10, 95% CI 1.05–1.15, *p* < 0.001), higher depressive symptoms (OR= 1.12, 95% CI 1.07–1.17, *p* < 0.001) and higher number of late effects (*p* < 0.001) were associated with low satisfaction with information received about late effects (Table 2). In the multivariable model, only female sex (OR = 0.47 comparing males to females, 95% CI 0.32–0.73, *p* = 0.001), lower health literacy levels (OR = 0.96, 95% CI 0.94–0.99, *p* = 0.008) and higher number of late effects (OR = 4.17, 95% CI 2.16–8.08, comparing 5 to 0 late effects, *p* = 0.038) were associated with lower satisfaction of information received. There were no significant interactions (no *p* < 0.01) between type of survivor (CCS vs. YACS) and any of the factors investigated (Table 2).

Aim 3a: Knowledge about late effects by group of survivors.

There was great variation in the proportion of survivors reporting knowledge of the 17 different late effects (Appendix A). Most survivors had knowledge of reduced fertility, new cancers, fatigue and psychological reactions, whereas few had knowledge of neurological pain and hearing problems. More CCS knew (knew or had experienced) about cardiovascular late effects and dental and hearing problems compared to YACS (all *p*-values <0.001). On the other hand, YACS showed better knowledge of hormonal changes, fatigue, sexual functioning, osteoporosis, lymphedema, psychological consequences and consequences of radiation (all *p*-values < 0.001) (Appendix A). 

Aim 3b: Factors associated with low knowledge about late effects.

In the univariable regression models, we found the following factors to be significantly associated with lower knowledge about all four late effects of fatigue, second cancers, hormonal changes and cognitive impairment: male sex (*p* < 0.001, *p* < 0.001, *p* < 0.001 and *p* = 0.004, respectively), lower education (all *p* < 0.001) and lower health literacy (*p* = 0.009, *p* < 0.001, *p* < 0.001 and *p* < 0.001, respectively; Table 3). In addition, cancer diagnosis, treatment and number of diagnoses were associated with all four outcomes (all *p* < 0.003), although in different directions depending on the late effect. Further, being a CCS compared to YACS was associated with lower knowledge about fatigue (*p* = 0.002) and hormonal changes (*p* > 0.001); lower age at diagnosis was associated with lower knowledge of fatigue, second cancers and hormonal changes (all *p* < 0.001); longer time since diagnosis was associated with lower knowledge about fatigue (*p* < 0.001), hormonal changes (*p* = 0.006) and cognitive impairment (*p* = 0.006); and earlier year of diagnosis was associated with lower knowledge about fatigue (*p* < 0.001) and second cancers (*p* = 0.22). 

In the univariable regression models, the following factors were significantly associated with lower knowledge about all four late effects of fatigue, second cancers, hormonal changes and cognitive impairment: male sex (all *p* < 0.001), lower education (*p* < 0.001, *p* = 0.004, *p* = 0.033 and *p* = 0.019, respectively) and lower health literacy (*p* = 0.038, *p* = 0.019, *p* = 0.001 and *p* = 0.001, respectively; Table 4). The type of cancer survivor was not associated and the diagnostic treatment received was only associated with knowledge about second cancers (*p* < 0.001). A lower number of late effects was associated with lower knowledge about fatigue, second cancers and hormonal changes (all *p* < 0.001) but higher knowledge about cognitive impairment (*p* = 0.013). There was no association with anxiety; however, survivors with higher depression scores had lower knowledge of hormonal changes (*p* = 0.022) and cognitive impairment (*p* < 0.001). 

## 4. Discussion

This study found that a substantial number of CCS and YACS reported not having received written information about their cancer history or any information about late effects. This was reported across survivor groups and type of treatment received. Those who had received information about late effects were most satisfied with the information provided by doctors and nurses at the hospital compared to the cancer nurse at municipality, General Practitioner GP, cancer society and other patient organizations. We also found that a high proportion of survivors still do not have knowledge about important late effects such as fatigue, second cancers, hormonal changes and cognitive impairment, with male sex, lower education, intensity of treatment and health literacy being the most prominent associated factors. 

### 4.1. Comparison with Other Studies

The present study corroborates the results of other research conducted in the same and other cancer populations [21,24,27,48]. Although our sample represents very long-term survivors treated in different “clinical eras”, there is reason to believe that there is still a need to improve information provision, especially targeted and systematic handover of information, in current clinical practice [12,49,50,51]. When looking at satisfaction with the source of information, a study from Great Britain, similarly to the present survey, found that survivors consider the treating hospital to be the best place to be advised about health [21]. Further, when looking at knowledge about late effects, male survivors and survivors with lower levels of education showed lower knowledge of all types of late effects investigated [30]. In line with other studies, we also found that better health literacy plays an important role in the satisfaction with information and survivors’ knowledge [52,53]. 

### 4.2. Interpretation of Results and Implications for Practice

CCS and YACS represent special groups, not only from a biological point of view, but also because of the phase of life in which they occur. Regarding information provision in CCS and YACS, health providers face the challenge of reaching the patients and not only the parents [54]. Because of the young age at diagnosis, patients and parents may not remember the details about the cancer history and the recommendations that may become relevant many years later. Therefore, providing patients and parents with a written treatment summary or information about their risks of late effects and recommended follow-up, akin to survivorship passports, is recommended, at least in pediatric oncology [55,56,57]. Additionally, our survivors were diagnosed from 1985, and patient procedures have evolved since then, with more focus on patient information and empowerment in recent years. Another challenge is the loss of information that the process of transition from pediatric to adult care may entail for CCS [58]. In addition, young adult cancer survivors can be a difficult group to target because many want to “move on” with their lives, tackle other important challenges (starting a family, a career) and tend to adhere less to survivorship recommendations [16]. 

We found that survivors were more satisfied with the information provided within the specialized hospital setting than from other information sources. Whether this reflects higher trust in clinicians at the treating hospital compared to others or a need for training of actors operating outside the specialized hospital cannot be established from our data. 

Early detection and prevention of late effects plays a central role in survivorship management and is an important part of the information provision process. Studies have shown that knowledgeable patients make better lifestyle choices and have better quality of life and less psychological distress [12,59]. We found high variation in the knowledge about the different late effects. For example, about half of survivors reported not to know about cardiovascular sequelae, which have significant impacts on survivors’ long-term health [5]. Factors associated with low knowledge across types of late effects were of male sex and low education. Health literacy seems to play an important, protective role, suggesting that it is necessary to consider survivors’ health literacy levels and to provide tailored information [35]. 

Although health care professionals need to educate the survivors on their risks of late effects and positive lifestyle choices to mitigate these risks, communicating such information can be challenging. Written information should be provided to make sure survivors have documents to hold on to. This should be especially true for these survivors diagnosed at a very young age. Systematic routines and approaches to provide this information, e.g., survivorship passports, may act as a communication tool for clinicians [60]. The latter is often cited as one solution, although its efficacy has yet to be established [61]. Improving health literacy skills and tailoring information, especially among adolescents, could also be an intervention approach worth considering [35]. Making information understandable for this group of patients and considering their individual characteristics could help overcome the communication barriers that adolescents and young adults often face [62]. Finally, identifying factors associated with low satisfaction with information and low knowledge about late effects and understanding the differences between CCS and YACS can help guide medical personnel to tailor information provision to CAYACS. 

### 4.3. Strengths and Limitations

A major strength of our study is the nation-wide, population-based cohort consisting of 5-year survivors of several diagnostic groups of childhood, adolescent and young adult cancers for whom basic demographic and clinical data are available from the CRN. 

A limitation is the low response rate, which might affect the sample representativeness. We did, however, estimate the effect of non-response bias in this cohort [39] and did not find evidence for strong bias on a wider range of survey outcomes. Unfortunately, in this study, due to the lack of detailed information about treatment, it was impossible to create risk groups, which would have allowed a better understanding of survivors’ knowledge about late effects.

## 5. Conclusions

Given the long life expectancy of CAYACs, knowledge of their own medical history and risk of late effects is especially important. Moreover, informed patients are more likely to adhere to follow-up care and less prone to psychological distress and lower quality of life. Thus, the large knowledge gaps uncovered in this study call for efforts to find ways to provide information not only to survivors engaged in follow-up care programs, but also those not engaged in any formal survivorship care. How to achieve this is complex, depending on local health care systems and opportunities to identify and contact survivors. However, such information is important for the survivors to understand their health conditions (including late effects), seek help when needed and to engage in meaningful self-management in order to improve the long-term care and health for young cancer survivors.

## Figures and Tables

**Figure 1 cancers-14-01534-f001:**
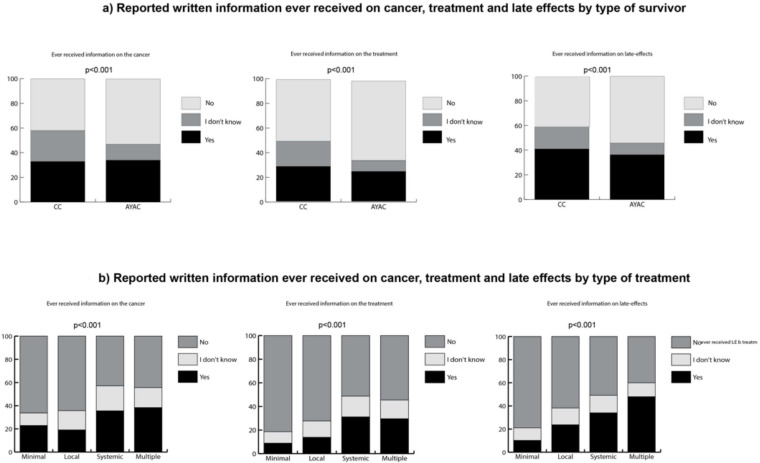
Reported written information ever received on cancer, treatment and late effects.

**Table 1 cancers-14-01534-t001:** Characteristics of the study sample.

	Childhood Cancer		Young Adult Cancer		*p*-Value ^a^
	(n = 607)		(n = 1411)		
Sociodemographic factors					
	N	% ^b^	N	% ^b^	
Sex					<0.001
Male	262	43	377	27	
Female	345	57	1034	73	
Education					0.065
Mandatory school or less	19	3	63	4	
High school	253	42	518	37	
University college/University	333	55	822	59	
Cancer related factors					
Childhood cancer diagnoses			na	na	na
Leukemia	192	32			
Lymphoma	148	24			
Neuroblastoma	11	2			
Retinoblastoma	13	2			
Renal tumor	30	5			
Hepatic tumor	13	2			
Bone tumors	48	8			
Short Tissue Sarcoma	27	5			
Germ cell tumor	69	11			
Other ^c^	55	9			
AYA cancer diagnoses	na	na			na
Colorectal cancer			149	11	
Breast cancer			557	39	
Melanoma			311	22	
Lymphoma			242	17	
Leukemia			152	11	
Treatment					<0.001
Minimal (melanoma)	0	0	273	21	
Local treatment	70	12	133	10	
Systemic single	162	27	144	11	
Multiple treatment	364	61	759	58	
Number of cancer diseases					0.001
1	596	97	1350	96	
2 (or more)	11	3	61	4	
Number of late effects					
0	228	37	521	37	0.001
01-Feb	159	26	302	21	
03-Apr	118	20	246	17	
>5	102	17	342	24	
Sociodemographic factors					
	Mean	SD	Mean	SD	*p*-value ^d^
Age at study	30.2	7.9	48.9	7.8	na
Age at diagnosis	10.6	5.9	32.9	5.3	na
Time since diagnosis	19.1	6.6	15.4	6.1	<0.001
Health literacy score	35.2	7.4	35	7.3	0.531
Anxiety (HADS)	5.3	4.1	4.6	3.8	0.001
Minor depression (PHQ9)	6.1	5.2	5.3	4.7	0.002
Quality of life (PCS)	49.3	9.8	47.9	10.1	0.006
Quality of life (MCS)	46.6	11.2	48.7	10.8	0.001
	Median	Range	Median	Range	
Year of diagnosis	1995	1985–2009	2000	1985–2009	

NOTE: ^a^
*p*-Values calculated from Chi-square Childhood Cancer Survivors with Adolescent and Young Adult Cancer survivors. ^b^ Column percentages are given. ^c^ Other. ^d^ *p*-Values calculated based on two-sample mean-comparison test (*t*-test). Abbreviations: na, not applicable.

**Table 2 cancers-14-01534-t002:** Factors associated with not being satisfied with the information received by a medical professional in univariable and multivariable logistic regression models.

	Univariable Regression (n = 644)				Multivariable Regression (n = 644)				P_Interaction_ ^b^
	OR	95% CI		*p*-Value ^a^	OR	95% CI		*p*-Value ^a^	
Age at study (years)	0.99	0.98	1.01	0.597	0.97	0.94	1.01	0.157	0.027
Sex								0.001	0.663
Female	ref			<0.001	ref				
Male	0.44	0.29	0.65		0.47	0.32	0.73		
Type of survivor									
CCS	ref			0.931	ref			0.413	
YACS	1.02	0.72	1.43		1.41	0.62	3.17		
Education				0.274					0.679
Mandatory school/less	0.59	0.23	1.52		0.52	0.18	1.39	0.187	
High School	0.48	0.19	1.23		0.47	0.17	1.27		
University	ref				ref				
Health Literacy	0.94	0.92	0.97	<0.001	0.96	0.94	0.99	0.008	0.412
Anxiety (HADS)	1.10	1.05	1.15	<0.001	0.70	−1.03	2.43		0.178
Minor Depression (PHQ9)	1.12	1.07	1.17	<0.001	1.03	0.98	1.09	0.186	0.907
Diagnosis ^c^				0.304					
Childhood cancer	ref				na	na	na	na	
Melanoma	0.52	0.21	1.35		na	na	na	na	
Breast cancer	1.28	0.85	1.93		na	na	na	na	
Colorectal cancer	0.88	0.39	1.92		na	na	na	na	
Non-Hodgkin Lymphoma	0.91	0.55	1.51		na	na	na	na	
LEUK	0.78	0.41	1.49		na	na	na	na	
Treatment				0.390				0.421	0.865
Minimal treatment	ref				ref				
Local treatment	1.68	0.51	5.49		1.69	0.47	6.08		
Systemic single ttm	1.52	0.51	4.45		1.14	0.35	3.68		
Multiple treatments	2.22	0.81	5.97		1.11	0.36	3.34		
Number of cancers				0.970				0.750	0.865
1	ref				ref				
>1	0.98	0.42	2.31		0.86	0.38	2.19		
Number of late effects				<0.001				0.038	0.799
0	ref				ref				
1–2	2.16	1.17	3.97		1.95	1.04	3.69		
3–4	3.84	2.12	6.99		3.37	1.76	6.48		
5	5.33	3.01	9.45		4.17	2.16	8.01		
Age at diagnosis (years) ^d^	0.99	0.98	1.02	0.853	na	na	na	na	
Time since first diagnosis	0.99	0.97	1.01	0.524	1.02	0.98	1.07	0.169	0.055
Year of diagnosis ^d^	1.01	0.98	103	0.565	na	na	na	na.	

Abbreviations: ref, reference; CI, confidence interval; OR, odds ratio; LEUK, leukemias; n.a., not applicable; ^a^
*p*-value from logistic regression models; ^b^ *p*-value for interaction between the respective variable with survivor type; ^c^ diagnostic group was not included in the multivariable model due to collinearity with type of survivor; ^d^ age at diagnosis and year of diagnosis were not included in the multivariable model due to collinearity with age and time since diagnosis.

**Table 3 cancers-14-01534-t003:** Factors associated with low knowledge about different late effects from univariable logistic regression models (n = 1911).

	Low Knowledge about Fatigue				Low Knowledge about Second Cancers					Low Knowledge about Hormonal Changes				Low Knowledge about Cognitive Impairment		
	OR	95% CI		*p*-Value ^a^	OR	95% CI		*p*-Value ^a^	OR	95% CI		*p*-Value ^a^	OR	95% CI		*p*-Value ^a^
Sociodemographic characteristics																
Age at study	0.99	0.98	1.01	0.931	1.01	1.00	1.03	0.003	0.98	0.97	0.99	<0.001	0.99	0.98	1.01	0.849
Type of survivor				0.002				0.228				<0.001				<0.001
CCS	ref				ref				ref				ref			
YACS	0.70	0.57	0.88		1.14	0.91	1.43		0.61	0.51	0.76		1.01	0.82	1.23	
Sex				<0.001				<0.001				<0.001				0.004
Female	ref				ref				ref				ref			
Male	2.81	2.31	3.56		1.71	1.38	2.12		2.86	2.34	3.51		1.35	1.01	1.41	
Education				<0.001				<0.001				<0.001				<0.001
Mandatory school or less	ref				ref				ref				ref			
High School	0.71	0.44	1.15		0.61	0.37	0.97		0.62	0.39	1.00		0.49	0.27	0.92	
University college/University	0.36	0.22	0.58		0.40	0.25	0.65		0.40	0.25	0.64		0.32	0.17	0.59	
Health Literacy	0.98	0.97	0.99	0.009	0.98	0.96	0.98	<0.001	0.97	0.96	0.98	<0.001	0.95	0.94	0.97	<0.001
Anxiety	0.95	0.50	0.76	0.078	1.02	0.99	1.04	0.127	1.02	0.99	1.04	0.082	1.02	0.99	1.04	0.082
Minor Depression	0.76	0.95	0.99	0.003	1.01	0.99	1.03	0.123	1.01	0.99	1.03	0.431	1.01	0.99	1.02	0.433
Cancer related factors																
Diagnosis				<0.001				0.001				0.001				0.001
Childhood cancer	ref				ref				ref				ref			
Melanoma	1.38	1.11	2.05		0.96	0.68	1.34		1.07	0.80	1.42		0.64	0.48	0.86	
Breast cancer	0.27	0.28	0.51		1.09	0.83	1.44		0.31	0.23	0.41		1.22	0.95	1.57	
Colorectal cancer	1.23	0.77	1.71		2.22	1.45	3.29		1.30	0.89	1.90		1.40	0.93	2.12	
Non-Hodgkin Lymphoma	0.57	0.39	0.83		0.87	0.60	1.25		0.60	0.43	0.84		1.04	0.75	1.43	
LEUK	0.68	0.44	1.04		1.42	0.95	2.12		0.81	0.55	1.17		0.91	0.62	1.31	
Treatment				<0.001				0.002				<0.001				<0.001
Minimal treatment	ref				ref				ref				ref			
Local treatment	0.93	0.63	1.39		1.89	1.24	2.90		0.83	1.24	2.90		1.11	0.75	1.63	
Systemic single treatment	0.59	0.41	0.84		1.33	0.90	1.96		1.08	0.77	1.51		1.33	0.94	1.88	
Multiple treatments	0.36	0.26	0.48		1.02	0.74	1.42		0.48	0.37	0.65		1.83	1.38	2.43	
Number of late effects				<0.001				0.003				<0.001				<0.001
0	ref				ref				ref				ref			
1–2	0.49	0.39	0.65		0.68	0.52	0.90		0.64	0.51	0.82		0.95	0.75	1.21	
3–4	0.25	0.17	0.33		0.75	0.57	1.01		0.38	0.29	0.51		1.96	1.47	2.58	
5	0.47	0.03	0.09		0.56	0.42	0.75		0.17	0.15	0.24		4.16	3.08	5.61	
Number of cancers				0.217				0.928				<0.001				0.493
1	ref				ref				ref				ref			
>1	0.69	0.37	1.28		1.02	0.56	1.78		0.29	0.14	0.59		1.09	0.99	1.01	
Age at diagnosis (years)	0.98	0.97	0.99	<0.001	0.98	0.97	0.99	<0.001	0.98	0.97	0.99	<0.001	1.01	0.97	1.00	0.135
Time since first diagnosis	1.05	1.04	1.07	<0.001	1.01	0.99	1.02	0.112	1.02	1.00	1.03	0.006	1.02	1.00	1.03	0.006
Year of diagnosis	0.95	0.94	0.96	<0.001	0.98	0.97	0.99	0.022	0.98	0.96	0.99	0.074	1.01	0.99	1.02	0.139

Abbreviations: CI, confidence interval; CCS, childhood cancer survivors; LEUK, leukemias; OR, odds ratio; YACS, young adult cancer survivors; ^a^
*p*-values from logistic regression models.

**Table 4 cancers-14-01534-t004:** Factors associated with low knowledge about different late effects from multivariable logistic regression models (n = 1911).

		Low Knowledge about Fatigue				Low Knowledge about Second Cancers				Low Knowledge about Hormonal Changes				Low Knowledge about Cognitive Impairment			
		OR	95% CI		*p*-Value ^C^	OR	95% CI		*p*-Value ^c^	OR	95% CI		*p*-Value ^c^	OR	95% CI		*p*-Value ^c^
Sociodemographic characteristics																	
Age at study		0.99	0.97	1.01	0.601	1.02	0.99	1.04	0.063	0.98	0.96	0.99	0.023	1.01	0.98	1.02	0.949
Type of survivor^a^					0.633				0.914				0.267				0.768
CCS		ref				ref				ref				ref			
YACS		1.15	0.65	2.03		1.02	0.61	1.75		1.33	0.80	2.18		1.08	0.66	1.78	
Sex					<0.001				<0.001				<0.001				<0.001
Female		ref				ref				ref				ref			
Male		2.78	2.14	3.59		1.54	1.29	2.21		2.54	1.80	2.94		1.68	1.32	2.18	
Education					<0.001				0.004				0.033				0.019
Mandatory school or less		ref				ref				ref				ref			
High School		0.60	0.26	3.83		0.65	0.43	1.16		0.55	0.31	0.95		0.58	0.27	1.18	
University college/University		0.29	0.13	0.43		0.43	1.25	2.21		0.39	0.23	0.68		0.44	0.21	0.90	
Health Literacy		0.98	0.97	1.03	0.038	0.98	0.96	0.99	0.019	0.97	0.95	0.98	0.001	0.97	0.96	0.99	0.001
Anxiety		1.02	0.96	1.02	0.581	1.02	0.98	1.06	0.303	1.02	0.98	1.03	0.171	0.97	0.94	1.02	0.168
Minor Depression		1.03	0.99	1.07	0.113	1.03	0.99	1.05	0.239	1.03	1.01	1.07	0.022	1.11	1.06	1.14	<0.001
Cancer related factors																	
Diagnosis^b^																	
Childhood cancer		na	na	na		na	na	na		na	na	na		na	na	na	
Melanoma		na	na	na		na	na	na		na	na	na		na	na	na	
Breast cancer		na	na	na		na	na	na		na	na	na		na	na	na	
Colorectal cancer		na	na	na		na	na	na		na	na	na		na	na	na	
Non-Hodgkin Lymphoma		na	na	na		na	na	na		na	na	na		na	na	na	
LEUK																	
Treatment					0.179				<0.001				0.414				0.493
Minimal treatment		ref				ref				ref				ref			
Local treatment		1.02	0.64	1.61		2.43	1.54	3.98		0.83	0.53	1.29		1.06	0.69	1.63	
Systemic single treatment		0.66	0.41	1.03		1.71	1.14	2.81		1.01	0.67	1.53		0.86	0.57	1.31	
Multiple treatments		0.29	0.16	1.26		1.61	1.08	2.38		0.88	0.61	1.26		1.13	0.79	1.61	
Number of late effects					<0.001				<0.001				<0.001				0.013
	0	ref				ref				ref				ref			
	1–2	0.53	0.39	0.72		0.60	0.44	0.83		0.61	0.46	0.83		0.83	0.63	1.10	
	3–4	0.23	0.16	0.34		0.64	0.45	0.90		0.32	0.23	0.46		1.52	1.09	2.13	
	5	0.04	0.02	0.07		0.43	0.29	0.61		0.14	0.09	0.21		2.88	1.99	4.14	
Number of cancers					0.426				0.821				0.090				0.969
	1	ref				ref				ref				ref			
	>1	1.35	0.64	2.85		1.07	0.56	2.02		0.51	0.24	1.01		1.01	0.55	1.87	
Age at diagnosis (years)		na	na	na	na	na	na	na		na	na	na		na	na	na	
Time since first diagnosis		1.04	1.02	1.07	0.002	0.99	0.96	1.02	0.735	1.02	1.06		0.174	0.99		1.02	0.862
Year of diagnosis		na	na	na	na	na	na	na		na	na			na		na	

Abbreviations: CI, confidence interval; CCS, childhood cancer survivors; LEUK, leukemias; na, not applicable; OR, odds ratio; YACS, young adult cancer survivors; ^a^ diagnostic group was not included in the multivariable model due to collinearity with type of survivor; ^b^ age at diagnosis and year of diagnosis were not included in the multivariable model due to collinearity with age and time since diagnosis; ^c^ global *p*-value calculated with likelihood ratio test.

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
