# Peer review of "Who Knows? Information Received, and Knowledge about, Cancer, Treatment and Late Effects in a National Cohort of Long-Term Childhood, Adolescent and Young Adult Cancer Survivors"

_cancers, 2022, doi:10.3390/cancers14061534_

Round 1

Reviewer 1 Report

This article presents the analysis of information received and knowledge about cancer and its late effects among childhood, adolescent and young adult survivors in Norway. The large sample is taken from a subset of a comprehensive national cancer registry, and the aims and results of the study are clearly presented.

From a statistical point of view, most of the methods used are simple and appropriate. However, I think there are two areas in which major improvements should be made:

  1. Given that the eligible population is drawn from a national cancer registry, for which some information (at least gender, date of birth, diagnosis, date of diagnosis, and number of cancers) is known, there is a good opportunity to reduce the effect of non-response bias by using that information. Reference 39, cited in the discussion,  suggests that the bias might not be particularly large for some questions related to information needs, but I don't think that's strong enough justification for effectively assuming that it is zero in this paper when it is possible to do better. In particular, there are differences in response rates associated with gender and age, both of which were found to be associated with satisfaction with information in this paper, hinting at the possibility of non-negligible bias.
  2. "Univariable prefiltering" prior to fitting a multivariable model is problematic for a few reasons (see, for example, Sun et al 1996; doi 10.1016/0895-4356(96)00025-X), and severely biases the estimates, confidence intervals and p-values in the multivariable model. It would be better to fit a prespecified multivariable model including all covariates that you are interested in controlling for (excepting that you could only include two of the four variables related to age and time of diagnosis, since they are linearly related). This might clear up some confusion with the multivariable results, e.g. the fact that Education is listed as "na" in Table 4 but treatment is excluded from the table altogether. The analysis of interaction between group and covariates is reasonable, although rather than focusing interpretation on significance vs non-significance, given the difference in the size of the two groups (such that there is greater power in the YACS group for the same effect), it would be better to describe the differences in estimated magnitudes (see, for example, Gelman & Stern 2006; doi 10.1198/000313006X152649). It might be even further improved by doing this within the multivariable model, instead of using the univariable results as justification for fitting separate multivariable models, since the observed interactions might be an artefact of differential confounding in the univariable models.

There are also a number of other things that I think don't necessarily need to be changed, but would improve with some extra justification, clarification or caution in interpretation:

  1. "Knowledge" of a particular late effect was defined as a combination of responses "I know about it" and "I have experienced it myself". However, it could be plausible that one could have experienced a late effect without knowing or realising that it is potentially a late effect of treatment. This is particularly true for common ailments that might be unrelated to cancer (e.g. fatigue), and could partially explain higher rates of "Knowledge" among AYACS compared to CCS (e.g. for fatigue, where the "I know about it" proportion seems to be very similar). A consequence of this is that the factors associated with low "Knowledge" might be partially driven by factors that are associated with less risk of experiencing the late effects. Given the way the question was asked, it might not be possible to disentangle these effects, but I think stronger justification for using this definition is warranted, as well as more care in interpreting the results in light of this limitation.

  2. Why was the depression score taken from the PHQ-9, when the HADS also includes a measure of depression?

  3. Why were the number of late effects and the number of cancer diseases categorised into groups before analysis, and how were those groups chosen? Generally it is recommended to avoid this (see e.g. this post for 

    a list of reasons and references).

  4. Why were the age at study and time since diagnosis calculated as of 15 May 2015? In any case, I would expect the age at study to be the age at diagnosis plus the time since diagnosis for each individual, and so the within-group means should also have that relationship. But that doesn't seem to be the case with the figures given in Table 1. Why is that? Also, given the definition of CCSs and AYACSs, we would not expect the mean ages at diagnosis to be equal between the two groups, so the hypothesis test in Table 1 is not very meaningful; the same applies to the age at study, given the inclusion criterion that restricts the year of diagnosis.

  5. Supplementary Figure 1 explains why only 2018 of the 2104 respondents were included in the analysis, but it would be good to include this explanation in the main text.

  6. It seems like similar percentages of CCSs and AYACSs report receiving information, and the difference in the rates of reporting not having received information is mainly driven by a larger proportion of CCSs responding "I don't know". This seems to be hinted at in the discussion and recommendation to provide written information, but could be addressed more explicitly.

  7. It's unclear whether satisfaction was asked separately for each information provider, or if it was just a single question. If it was a single question, then the analyses that are stratified by provider do not provide a simple interpretation, since satisfaction associated with a particular provider may be reflecting information received from other providers, and results are dependent on the differing combinations of providers, which might differ between survivors. At the least, this should be supplemented with information about distribution of information providers across the two groups. If it was separate questions, then the interpretation is simpler, but it isn't clear what definition was used for the logistic regression models (e.g. best response across all providers?).

  8. In some places (e.g. P6 L216-217), a non-significant p-value is interpreted as "no difference". These should be reworded, and in general rather than referring only to p-values, it would be good to include the magnitude of the plausible differences (i.e. confidence intervals) to aid interpretation throughout the paper. The relatively large sample size means that you could have high power for differences that are not clinically meaningful.

And some more minor things:

  1. The hypothesis tests in Table 1 compare the characteristics of CCS vs YACS respondents, and we wouldn't expect the null hypothesis to be true unless the characteristics in the population were equal between groups (or the unlikely scenario that the differential response exactly offsets the population differences). The title of Section 3.1 could be changed to "Characteristics of the study sample" and the text could be worded more accurately -- e.g. "In comparison with YACS respondents, more CCS in our sample were males…".

  2. I think it would be better to display the "I don't know" category in between "Yes" and "No" in Figure 1, so that the "Yes" and "No" percentages can be more easily compared between groups visually.

  3. The numbers in the first paragraph of P7 don't seem to agree with what is presented in Figure 2. e.g. the text says "specialist physicians in the hospital (67% CCS and 75% YACS) and cancer nurses at the hospital (70% CCS and 66% YACS)", but in Figure 2, all of these percentages are at least 80%. Similarly, the text says "general practitioners (53% CCS and 41% YACS)" but in the figure, these percentages look to be 60% and around 65% respectively.

Finally some things that might be typos:

  1. Throughout - there is inconsistency in the use of "late effects" and "late-effects"

  2. P2 L89  - Should the inclusion criterion be ≥ 18 years, rather than ≤18?

  3. P2 L93 - Missing the LEUK abbreviation for leukaemia which is used later

  4. P5, Table 1 & P8 Table 2 - Strange symbol in the 5+ row of late effects

  5. P6, L208 - "p>0.001" instead of "p<0.001"

  6. P6 L217 - "0.061<p-value>0.566" instead of "0.061<p-value<0.566"

  7. P7 Figure 2 - "Othe patient organization" in bar label

  8. P7 L224 & P8 L235 & P9 L243 - "information received by a medical professional"; should these say "...from a medical professional"?

  9. P8 Table 2 - HADS and PHQ9 are listed under "sociodemographic characteristics", but in Table 4 they are under "psychological health"

  10. P9 Table 4 - CI for male YACS seems to be copied from the age row; it doesn't include the point estimate

  11. P10 L276 - "all ps>0.001" instead of "all ps<0.001" (twice)? Except e.g. treatment for second cancers is p = 0.002

  12. P14 L323 - "are less susceptible to adhere less" is a little unclear; could it be reworded?

  13. Supplemental Table 1 seems to be a copy of Table 3 in the main paper

Reviewer 2 Report

The text by Gianinazzi et al. describes a large series of patients treated for neoplastic disease in pediatric or young adults. The authors ask surviving patients a questionnaire about the written information provided on disease and treatment, patient satisfaction with the information received and the side effects related to the therapy.

1) the questionnaire administered to the opazienti must be proposed among the materials and methods

2) The authors declare that they cannot stratify patients based on the treatment received and this should also be indicated in the abstract.

3) Figure 2 can be removed because it is already reported in the text

4) In Tables 2 and 5 the part related to the treatment can be removed. 

Author Response

The questionnaire administered to the patients must be proposed among the materials and methods

This is a valid suggestion. We talked about it previously and we decided not to include the questionnaire in the supplemental material because it is only available in Norwegian.

The authors declare that they cannot stratify patients based on the treatment received and this should also be indicated in the abstract.

Thank you for suggesting this. We added the requested sentence in the abstract.

Figure 2 can be removed because it is already reported in the text.

We agree with the reviewer in that Figure 2 is redundant and does not add any information. We eliminated it.

In Tables 2 and 5 the part related to the treatment can be removed

We cannot remove the treatment part because of Reviewer 1’s comments and because of the way we chose the variables for the models.

Reviewer 3 Report

My compliments to the authors for their attention about a very important (but  for a long time ignored) topic. 

Just as a curiosity, in Italy the issue is so felt that  the scientific societies of pediatric oncohematology and adult oncology have created a task force for AYA patients (https://pubmed.ncbi.nlm.nih.gov/34841968/)

Returning to the paper, some minor revisions:

  • patient sample: as you have correctly reported, the low response rate is certainly a weakness of the study. Did you plan in advance a minimum response rate to consider the results worthy of publication? If yes: please details in methods. If no...why?
  • You have devoted ample space to the results but little to discussion and conclusions. What do you think is the reason for the findings? And above all, do you have any future ideas / proposals? To be detailed

Author Response

Just as a curiosity, in Italy the issue is so felt that the scientific societies of pediatric onco-hematology and adult oncology have created a task force for AYA patients (https://pubmed.ncbi.nlm.nih.gov/34841968/)

Thank you for bringing our attention to this fantastic effort! We should follow suit in Norway too!

Returning to the paper, some minor revisions:

Patient sample: as you have correctly reported, the low response rate is certainly a weakness of the study. Did you plan in advance a minimum response rate to consider the results worthy of publication? If yes: please details in methods. If no...why?

This is a great question. The short answer is no - it was not planed in advanced for a low response rate. The response rate was considerably lower than expected based on other survey studies done by our group (around 60-70%). However, once we had collected the data, and became aware of the low response rate, we did a literature search on response rates to population-based studies. In this literature- the response rate has decreased dramatically in the past two decades and a response rate of 30-40 % is the norm rather than the exception. Our sample was recruited from a nation-wide population of long-term cancer survivors who would not have been in regular follow up because of the cancer for years or decades, and could thus be more comparable to general population based samples compared to survivor samples recruited through ongoing studies or medical institutions. Luckily, we had some data available for the whole population from the cancer registry, enabling us to model potential impact of non-response bias ((Lie, H.C., Rueegg, C.S., Fosså, S.D. et al. Limited evidence of non-response bias despite modest response rate in a nationwide survey of long-term cancer survivors—results from the NOR-CAYACS study. J Cancer Surviv 13, 353–363 (2019). https://doi.org/10.1007/s11764-019-00757-x). We found very low risk of non-response bias in a range of patient related outcomes and therefore consider the risk of publishing on the material for low.

You have devoted ample space to the results but little to discussion and conclusions. What do you think is the reason for the findings? And above all, do you have any future ideas / proposals? To be detailed

We now expanded the conclusion part in the discussion.

“Given the long life expectancy of CAYACs, knowledge of their own medical history and risk of late effects is especially important. Moreover, informed patients are more likely to adhere to follow-up care and less prone to psychological distress and lower quality of life. Thus the large knowledge gaps uncovered in this study call for efforts to find ways to provide information not only to survivors engaged in follow-up care programs, but also those not engaged in any formal survivorship care. How to achieve this is complex, depending on local health care systems and opportunities to identify and contact survivors. Yet, such information is important for the survivors to understand their health conditions (including late effects), seek help when needed, and to engage in meaningful self-management in order to improve the long term care and health for young cancer survivors.”

Round 2

Reviewer 1 Report

I'd like to thank the authors for their thorough and thoughtful responses. I think the analysis of satisfaction is much improved and clearer.

But it isn't clear to me why univariable prefiltering is still used for the analyses of knowledge about late effects: the justification that there is "little knowledge on these outcomes in the literature" doesn't mean that the statistical issues can be ignored, and doesn't seem to be consistent with the fact that these models have excluded some variables (survivor type, anxiety, depression) that were explored in the satisfaction model.

In the knowledge models, there is still the issue of linear relationships between the age and time variables, which is not just a problem of collinearity but also of interpretation (the odds ratios from the multivariable models represent the change in probability of the outcome that is associated with with an increase of 1 year in the variable of interest, keeping all of the others constant, which is impossible given their deterministic relationships). There also seem to be some errors in implementation, e.g. year of diagnosis has p = 0.02 in the univariable model for second cancers, and time since first diagnosis has p = 0.006 in the univariable model for hormonal changes, but neither are included in the respective multivariable models. Three of the four age/time variables are included in the multivariable model for fatigue.

I'm sorry that this suggestion would mean some more work but I think it is important to avoid using statistical methods with bad properties in the literature.

A couple of more minor things:

  • Age at study and age at diagnosis would be expected to be different between CCS and YACS groups by definition, so a test of the null hypothesis that they are the same isn't meaningful. So I would suggest removing these p-values from Table 1.
  • P5: I think that "survivors aged 18 years at the time of the study" should be " 18 years", which would agree with the fact that the mean age at study are > 18 years in both groups in Table 1. If I have misunderstood, perhaps the reason for this apparent discrepancy should be explained.
  • P9: "0.061<p-value>0.566" should be "0.061<p-value<0.566"

Author Response

But it isn't clear to me why univariable prefiltering is still used for the analyses of knowledge about late effects: the justification that there is "little knowledge on these outcomes in the literature" doesn't mean that the statistical issues can be ignored, and doesn't seem to be consistent with the fact that these models have excluded some variables (survivor type, anxiety, depression) that were explored in the satisfaction model.

In the knowledge models, there is still the issue of linear relationships between the age and time variables, which is not just a problem of collinearity but also of interpretation (the odds ratios from the multivariable models represent the change in probability of the outcome that is associated with with an increase of 1 year in the variable of interest, keeping all of the others constant, which is impossible given their deterministic relationships). There also seem to be some errors in implementation, e.g. year of diagnosis has p = 0.02 in the univariable model for second cancers, and time since first diagnosis has p = 0.006 in the univariable model for hormonal changes, but neither are included in the respective multivariable models. Three of the four age/time variables are included in the multivariable model for fatigue.

I'm sorry that this suggestion would mean some more work but I think it is important to avoid using statistical methods with bad properties in the literature.

The reviewer is right. We changed all the models (see table 3 and table 4) accordingly.

A couple of more minor things:

  • Age at study and age at diagnosis would be expected to be different between CCS and YACS groups by definition, so a test of the null hypothesis that they are the same isn't meaningful. So I would suggest removing these p-values from Table 1.

We changed this as suggested by the reviewer

  • P5: I think that "survivors aged ≤ 18 years at the time of the study" should be "≥ 18 years", which would agree with the fact that the mean age at study are > 18 years in both groups in Table 1. If I have misunderstood, perhaps the reason for this apparent discrepancy should be explained.

We changed this as suggested by the reviewer

Reviewer 2 Report

Authors full replied to all criticisms. No other questions.

Author Response

Ok.